# Estimation of PM$_{2.5}$ Concentration in China Using Linear Hybrid Machine Learning Model

Zhihao Song[1], Bin Chen[1], Yue Huang[1], Li Dong[1], Tingting Yang[2]

[1]Atmospheric Science College of Lanzhou University, Lanzhou 730000, China

[2]Gansu Seed General Station, Lanzhou 730030, China

*Correspondence to*: Bin Chen (chenbin@lzu.edu.cn)

**Abstract.** The satellite remote-sensing aerosol optical depth (AOD) and meteorological elements were employed to invert PM$_{2.5}$ (The fine particulate matter with a diameter below 2.5μm) in order to control air pollution more effectively. This paper proposes a restricted gradient-descent linear hybrid machine learning model (RGD–LHMLM) by integrating a random forest (RF), a gradient boosting regression tree (GBRT), and a deep neural network (DNN) to estimate the concentration of PM$_{2.5}$ in China in 2019. The research data included Himawari-8 AOD with high spatiotemporal resolution, ERA-5 meteorological data, and geographic information. The results showed that, in the hybrid model developed by linear fitting, the DNN accounted for the largest proportion, whereas the weight coefficient was 0.62. The R$^2$ values of RF, GBRT, and DNN were reported 0.79, 0.81, and 0.8, respectively. Preferably, the generalization ability of the mixed model was better than that of each sub-model, and R$^2$ (determination coefficient) reached 0.84, whereas RMSE (root mean square error) and MAE (mean Absolute Error) were reported 12.92 μg/m$^3$ and 8.01 μg/m$^3$, respectively. For the RGD-LHMLM, R$^2$ was above 0.7 in more than 70% of the sites, whereas RMSE and MAE were below 20 μg/m$^3$ and 15 μg/m$^3$, respectively, in more than 70% of the sites due to the correlation coefficient having seasonal difference between the meteorological factor and PM$_{2.5}$. Furthermore, the hybrid model performed best in winter (mean R$^2$ was 0.84) and worst in summer (mean R$^2$ was 0.71). The spatiotemporal distribution characteristics of PM$_{2.5}$ in China were then estimated and analyzed. According to the results, there was severe pollution in winter with an average concentration of PM$_{2.5}$ being reported 62.10 μg/m$^3$. However, there was slight pollution in summer with an average concentration of PM$_{2.5}$ being reported 47.39 μg/m$^3$. The period from 10:00 to 15:00 every day is the best time for model inversion, also at this time the pollution is high. The findings also indicate that North China and East China are more polluted than other areas and that their average annual concentration of PM$_{2.5}$ was reported 82.68 μg/m$^3$. Moreover, there was relatively low pollution in

Inner Mongolia, Qinghai, and Tibet, for their average $PM_{2.5}$ concentrations were reported below 40

$\mu g/m^3$.

**1 Background**

In recent years, pollutants have been discharged increasingly in China where air pollution is

becoming worse than ever before due to rapid urbanization and industrialization (Wang et al., 2019a).

The fine particulate matter ($PM_{2.5}$) with a diameter below 2.5μm is the main component of air pollutants

having considerable impacts on human health, atmospheric visibility, and climate change (Gao et al.,

2015; Pan et al., 2018; Pun et al., 2017; Qin et al., 2017). The global concern about $PM_{2.5}$ has increased

significantly since it was listed as a top carcinogen (Apte et al., 2015; Lim et al., 2020). Currently, ground

monitoring is the most efficient method of measuring $PM_{2.5}$ (Yang et al., 2018). However, monitoring

stations are not evenly distributed due to terrain and construction costs; therefore, it is difficult to obtain

a wide range of accurate $PM_{2.5}$ concentration data (Han et al., 2015). To solve the problem, the method

of estimating $PM_{2.5}$ with satellite remote-sensing was developed. Satellite remote-sensing is

characterized by a wide coverage and high resolution (Hoff and Christopher, 2009; Xu et al., 2021).

There is also a high correlation between AOD, obtained from satellite remote sensing inversion, and

$PM_{2.5}$; therefore, AOD is a very effective method of monitoring the spatiotemporal concentration

characteristics of $PM_{2.5}$.

After Engel-Cox et al. (2004) proposed using satellite AOD to estimate $PM_{2.5}$ concentration, several

studies are reported in the literature to address this theory. Based on the regression model, Liu et al. (2005)

introduced AOD, boundary layer height, relative humidity, and geographical parameters as the main

controlling factors to estimate $PM_{2.5}$ in the eastern part of the United States, and the verification

coefficient $R^2$ obtained was 0.46. Tian and Chen (2010) used AOD, $PM_{2.5}$, and meteorological parameters

in Southern Ontario, Canada, to establish a semi-empirical model to predict $PM_{2.5}$ concentration per hour,

and the verification coefficient $R^2$ obtained in rural and urban areas was 0.7 and 0.64, respectively. Hu et

al. (2013) proposed a geography weighted regression model to estimate the surface $PM_{2.5}$ concentration

in southeastern America by combining AOD, meteorological parameters, and land use information. Their

model average $R^2$ was 0.6. Lee et al. (2012) believed that the satellite remote sensing AOD data would

be interfered by clouds and snow and ice, and the reliability of the data was questionable. They proposed

a mixed model based on AOD calibration to predict the ground $PM_{2.5}$ concentration in New England, USA, and achieved good results ($R^2 = 0.83$). Li et al. (2016) used PMRS method to remote sensing ground $PM_{2.5}$. Combined with MODIS (Moderate-resolution Imaging Spectroradiometer) AOD and ground observation data, Lv et al. (2017) estimated the daily surface $PM_{2.5}$ concentration in the Beijing-Tianjin-Hebei region and improved the data resolution to 4 km. Using interpretable self-adaptive deep neural network, Chen et al. (2021) estimated daily spatially-continuous $PM_{2.5}$ concentrations across China, and analyzed the contribution of various characteristics to the $PM_{2.5}$ model. The data used in these early studies are AOD products obtained from polar-orbit satellite sensors. The daily observation frequency is limited. Due to the influence of cloud and ground reflection, the dynamic change information of $PM_{2.5}$ cannot be obtained. As a result, geostationary satellite observations can be used to overcome the problem of low temporal resolution for estimating surface $PM_{2.5}$ (Emili et al., 2010).

The Himawari-8 satellite commonly used in the Asia-Pacific region is a geostationary satellite launched by the Japan Meteorological Agency in 2014. The observation frequency is 10 minutes, and the observation results can characterize the aerosol and provide AOD data with a resolution of 5 km (Bessho et al., 2016; Yumimoto et al., 2016). Due to its excellent performance, Wei et al. (2021a) use Himawari-8 data to estimate ground $PM_{2.5}$, result shows that the CV-$R^2$ (cross-validation coefficient of determination) is 0.85, with a root-mean-square error (RMSE) and mean absolute error (MAE) of 13.62 and 8.49 $\mu g/m^3$, respectively. Wang et al. (2017) proposed an improved linear model, introduced AOD, meteorological parameters, geographic information to estimate $PM_{2.5}$ in the Beijing-Tianjin-Hebei region, and the verification coefficient $R^2$ was 0.86. Zhang et al. (2019b) used Himawari-8 hourly AOD product to estimate ground $PM_{2.5}$ in China's four major urban agglomerations. The results showed significant diurnal, seasonal, and spatial changes and improved the temporal resolution of estimating $PM_{2.5}$ concentration to the hourly level. Yin et al. (2021) used Himawari-8 hourly TOAR (top-of-the-atmosphere reflectance) data to estimate ground $PM_{2.5}$ in China, improved data coverage area.

As research into ground-based $PM_{2.5}$ estimation deepens, traditional linear or nonlinear models cannot meet the requirements of large-scale estimation and are gradually being replaced by machine learning algorithms with strong nonlinear fitting ability(Guo et al., 2021; Mao et al., 2021). Liu et al. (2018) combined Kriging interpolation and random forest algorithm to obtain the concentration of high-resolution ground $PM_{2.5}$ in the United States. To demonstrate the accuracy and superiority of the proposed method, the results were compared with the $PM_{2.5}$ concentration in ground measurement stations. Chen

et al. (2019) stacked and predicted PM$_{2.5}$ concentration based on a variety of machine learning algorithms, discussed the influence of meteorological factors on PM$_{2.5}$ and achieved an R$^2$ = 0.85. Li et al. (2017a) established a GRNN (Generalized regression neural networks) model for the whole of China to estimate PM$_{2.5}$ concentration, and the results demonstrated that the performance of the deep learning model was better than that of the traditional linear model. In addition, there are some novel algorithms such as space-time extra-trees (STET) (Wei et al., 2021b) and space-time random forest (STRF) (Wei et al., 2019a) that are also used for PM$_{2.5}$ inversion research.

A large number of existing studies in the broader literature have examined the estimation of ground PM$_{2.5}$ concentrations using satellite remote sensing AOD. However, the performance of PM$_{2.5}$ estimation models established in the existing studies varies greatly and the performance of the models is not stable in different seasons and regions. To overcome this limitation, in this paper, a linear hybrid machine learning model (RGD-LHMLM) based on random forest (RF), gradient lifting regression tree (GBRT), and deep neural network (DNN) is proposed to estimate ground PM$_{2.5}$ concentration. The model performance is evaluated from time and space to analyze its causes. Finally, spatiotemporal distribution of PM$_{2.5}$ concentration in China in 2019 is obtained.

**2 Data**

**2.1 Ground PM$_{2.5}$ Monitoring Data**

PM$_{2.5}$ concentration data for 2019 used in this study are available from the China Environmental Monitoring Center's Air Quality Real-Time Publication System. The PM$_{2.5}$ datasets are calibrated and quality-controlled according to national standards GB 3095-2012 (China's National Ambient air quality standards)(China, 2012).The system extracts hourly mean PM$_{2.5}$ data. By the end of 2019, China had 1641 monitoring stations built and in operation. Figure 1 shows the spatial distribution of monitoring stations in China.

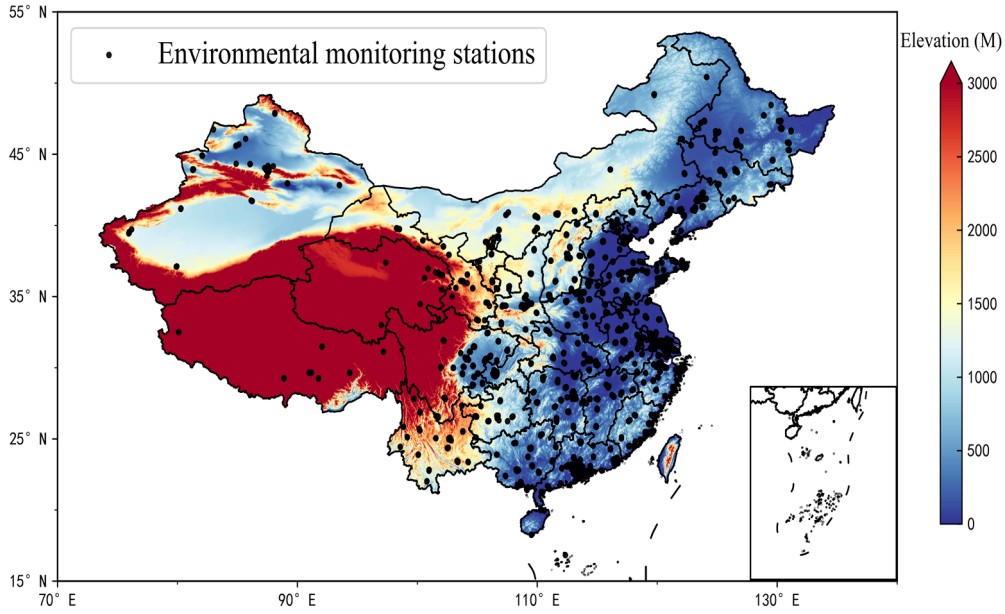

**Figure 1 Distribution diagram of Environmental monitoring stations in China**

**2.2 Satellite AOD Data**

The Advance Himawari Imager (AHI) on the Himawari-8 satellite launched by the Japan

Meteorological Agency is a highly improved multi-wavelength imager. It adopts the whole disk

observation method and has 16 visible and infrared channels. It has the characteristics of fast imaging

speed, flexible observation area, and time. Himawari-8 AOD is obtained by an aerosol retrieval algorithm

based on Lambertian-surface-assumed developed by Yoshida et al. (2018). The Level-3-hour AOD

product, released by the Japan Aerospace Space Agency (JAXA), provides 500 nm AOD data with a

spatial resolution of 5km during the day. In previous studies (Zang et al., 2018), Himawari-8 AOD was

compared with the AOD data of AERONET (Aerosol Robotic Network) in China and achieved good

performance (Zhang et al., 2019c), so that the results show that they are consistent ($R^2$=0.75), RMSE and

MAE were achieved 0.39 and 0.21, respectively(Wei et al., 2019b). The AOD data used in this study is

the Himawari-8 Level 3-hour AOD data in 2019 obtained from the Himawari Monitor website of the

Japan Meteorological Agency. In the study, we selected AOD with strict cloud screening, that is, AOD

data with low uncertainty.

**2.3 Meteorological Data**

ERA-5 reanalysis data is an hourly collection of atmospheric and land-surface meteorological

elements since 1979 that the European Centre (ECMWF) has used its prediction model and data

assimilation system to "Reanalyse" archived observations(Jiang et al., 2021). Data used in this paper

include surface relative humidity (RH, expressed as a percentage), air temperature at a height of 2 m

(TM, expressed as K), Wind speed (U10, V10, in m/s), surface pressure (SP, in Pa), boundary layer height

(BLH, in m) and cumulative precipitation (RAIN, in m) at 10 m above the ground. A series of studies

has indicated that these parameters can affect the concentration of $PM_{2.5}$ (Fang et al., 2016; Guo et al.,

2017; Li et al., 2017b; Wang et al., 2019b; Zheng et al., 2017; Gui et al., 2019). Uncertainty estimation

of ERA5 data has described in detail in the following website:

https://confluence.ecmwf.int/display/CKB/ERA5%3A+uncertainty+estimation.

**2.4 Auxiliary Data**

The auxiliary data used in this study include high and low vegetation index (LH, LL), ground

elevation data (DEM), and population density data (PD). The high and low vegetation index is derived

from ERA5 reanalysis data, which respectively represent half of the total green leaf area per unit level

ground area of high and low vegetation type. The ground elevation data are derived from SRTM-3

measurements jointly conducted by NASA and the Defense Department's National Mapping Agency

(NIMA), with a spatial resolution of 90 m. The population data come from the 2015 United Nations

Adjust Population Density data provided by NASA's Center for Socio-Economic Data and Applications

(SEDAC), which is based on national censuses and adjusted for relative spatial distribution.

**3 Method**

**3.1 Random Forest**

Random Forest (RF) is built based on the combination of the Bagging algorithm and decision

tree(Breiman, 2001), which is an extended variant of the parallel ensemble learning method (Stafoggia

et al., 2019). To construct a large number of decision trees, the random forest model takes multiple

samples of the sample data. In the decision tree, the nodes are divided into sub-nodes by using the

randomly selected optimal features until all the training samples of the node belong to the same class.

Finally, all the decision trees are merged to form the random forest. This method has proved to be

effective in regression and classification problems and is one of the most well-known Machine learning

algorithms used in many different fields (Yesilkanat, 2020).

**3.2 Gradient Boosted Regression Trees**

Different from the random forest, Gradient Boosting Regression Tree (GBRT) is based on Boosting algorithm and decision tree(Friedman, 2001). The basic principle of GBRT is to construct M different basic learners through multiple iterations, and constantly add the weight of the learners with a small error probability, to eventually generate a strong learner (Johnson et al., 2018). The core of this method is that after each iteration, a learner will be built in the direction of residual reduction (gradient direction) to make the residual decrease in the gradient direction (Schonlau, 2005). The basic learner of GBRT is the regression tree in the decision tree. During the prediction, a predicted value is calculated according to the model obtained. The minimum square root error is used to select the optimal feature to split the dataset, and the average value of the child node is then taken as the predicted value.

**3.3 Deep Neural Networks**

Deep Neural Networks (DNN) is a supervised learning technique that uses a backpropagation algorithm to minimize the loss function. It adjusts the parameters through an optimizer, and has high computational power, making it ideal for solving classification and regression problems (Wang and Sun, 2019). The structure of DNN includes an input layer, an output layer, and several hidden layers. Each layer takes the output of all nodes of the previous layer as the input, and this process requires activation functions. Compared with other activation functions, the linear rectifying function (ReLU) has the advantages of simple derivation, faster convergence, and higher efficiency. At the same time, among the adaptive learning rate optimizers, the Adamx optimizer performs the best. It not only has the advantages of Adam in determining the learning rate range and having stable parameters in each iteration but also simplifies the method of defining the upper limit range of the learning rate and improves the iteration efficiency (Diederik and Jimmy, 2015). Therefore, in this paper, we selected the Adamx optimizer and ReLU activation function to train the DNN.

**3.4 Model Establishment and Verification**

After data processing, RF, GBRT, and DNN are used for modeling.

$$PM_{2.5i,j} = AOD_{i,j} + BLH_{i,j} + RH_{i,j} + TM_{i,j} + LL_{i,j} + LH_{i,j} + SP_{i,j} \qquad (1)$$
$$+ RAIN_{i,j} + U_{10_{i,j}} + V_{10_{i,j}} + PD_{i,j} + HEIGHT_{i,j} + LON_{i,j} + LAT_{i,j}$$
$$+ MONTH_{i,j} + HOUR_{i,j}$$

Formula (1) is applicable to RF, GBRT and DNN, where $PM_{2.5i,j}$ is the $PM_{2.5}$ at time i on station j.

To prevent model parameters from being controlled by large or small range data and speed up the

convergence rate of the model, the data must be normalized before starting the training process. Finally,

the three optimal sub-models are linear combined to achieve the final mixed model. To verify the model

performance, this paper uses the "10-fold cross-validation" method (Adams et al., 2020). In this method,

the data is split into 10 copies, 9 copies for training and 1 copy for verification; this process is repeated

10 times, and then the average of the 10 predictions is computed as the final result. Finally, the predicted

value and the measured value are fitted linearly. At the same time, several indicators are used to evaluate

the model, including the mean absolute error (MAE, when the predicted value and the true value are

exactly equal to 0, that is, perfect model; The larger the error, the greater the value), the root mean square

error (RMSE, when the predicted value and the real value are completely consistent is equal to 0, that is,

the perfect model; The larger the error, the greater the value), the slope of the fitting equation and the

determination coefficient $R^2$ (the greater the value, the better the model fitting effect), the bias (Bias, is

the difference between the predicted values and the true values, so that models with larger bias performed

worse), and the GME (generalization error of the bias, It is generally believed that bias should be

expressed as a square when using generalization error). The calculation formula of each indicator is

shown as follows:

$$R^2 = 1 - \frac{ss_{res}}{ss_{tot}} \tag{2}$$

$$MAE = \frac{1}{n}\sum_{i=1}^{n}|\widehat{y_i} - y_i| \tag{3}$$

$$RMSE = \sqrt{\frac{1}{n}\sum_{i=1}^{n}(\widehat{y_i} - y_i)^2} \tag{4}$$

$$Bias = \frac{\sum_{i=1}^{N}\widehat{y_i} - y_i}{N} \tag{5}$$

$$GEB = \frac{\sum_{i=1}^{N}(\widehat{y_i} - y_i)^2}{N} \tag{6}$$

23     Where $\widehat{y_i}$ represents the predicted value, $y_i$ shows the true value, $ss_{res}$ denotes the error between

24     the regression data and the mean value, $SS_{tot}$ represents the error between the real data and the mean

25     value, and the mean value is the mean value of the true value.

26     The research process is illustrated in Figure 2:

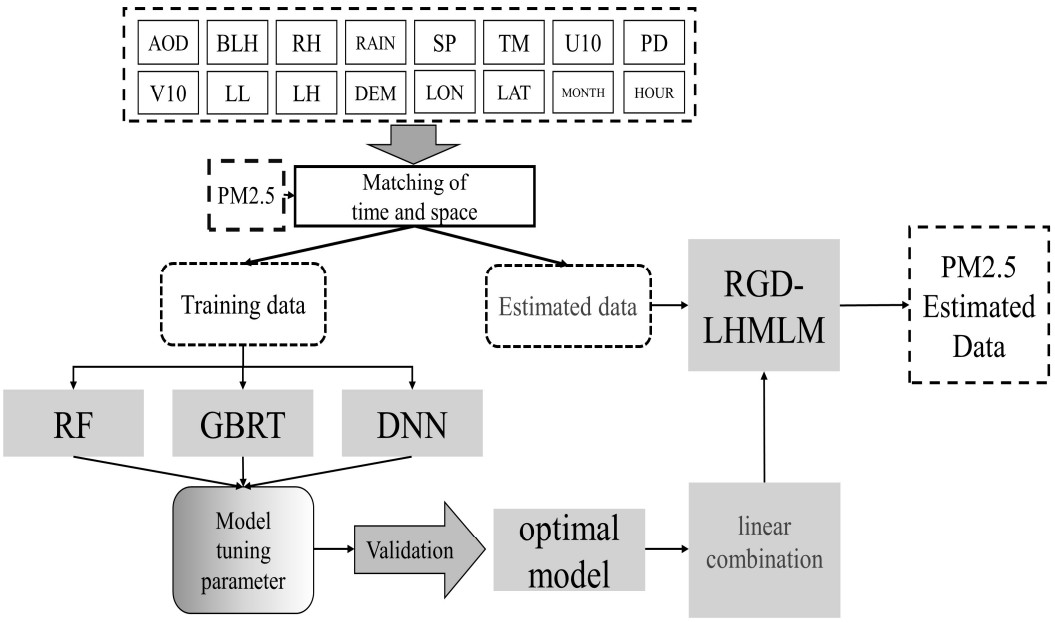

**Figure 2 Schematic diagram of model**

**4 Results and Discussion**

**4.1 Modeling Results**

According to the above steps, the mixed model RGD-LHMLM is obtained through modeling

verification, and is compared with RF, GBRT, and DNN. The fitting and verification accuracy results of

each model are shown in Table 1.

**Table 1 Comparison of model accuracy**

| Model | Fitting | | | | Validation | | | |
|---|---|---|---|---|---|---|---|---|
| | $R^2$ | RMSE | MAE | GEB | $R^2$ | RMSE | MAE | GEB |
| RF | 0.95 | 6.99 | 4.05 | 114.19 | 0.79 | 14.89 | 9.33 | 208.97 |
| GBRT | 0.96 | 6.87 | 4.52 | 110.00 | 0.81 | 14.09 | 9.18 | 198.65 |
| DNN | 0.97 | 5.03 | 3.49 | 59.16 | 0.80 | 14.45 | 9.06 | 221.86 |
| RGD-LHMLM | 0.98 | 4.39 | 3.00 | 44.97 | 0.84 | 12.92 | 8.01 | 166.95 |

9         The PM$_{2.5}$ inversion results of a single machine learning model show that DNN has the best

10   inversion performance, followed by GBRT, and RF has the worst performance. The expression of the

11   mixing model obtained after linear mixing is as follows:

12   $$PM_{2.5RGD-LHMLM} = 0.25PM_{2.5RF} + 0.17PM_{2.5GBRT} + 0.62PM_{2.5DNN} - 2.13 \qquad (7)$$

The weight coefficient of DNN in the mixed model was the largest (0.62). The $R^2$ of RGD-LHMLM in the training set was 0.98, and the RMSE was only 4.39 μg/m³, indicating that the model had an excellent data fitting effect. Meanwhile, the generalization ability of the mixed model is also good, with $R^2$ of 0.84 and RMSE of 12.92 μg/m³ on the validation data set. Among all the models, the deviation generalization error of the linear mixed model is also the lowest, indicating that the difference between the results obtained by this model and the real value is the least. Compared with RF, GBRT, and DNN, the inversion performance of RGD-LHMLM is improved. In other words, the combination of multiple models can improve the robustness and generalization ability of the model (Wolpert, 1992). The linear fitting equation coefficients between the predicted and measured values in the training set and the verification set were 0.98 and 0.84, respectively, indicating that the prediction accuracy of the model reached a high level. The fitting curve between the model predicted value and the real value is shown in Figure 3. The RGD-LHMLM model has the smallest degree of data dispersion, and the slope of the fitting line reaches 0.84, indicating that 84% of the prediction results are accurate, higher than the three sub-models. The accuracy of the model decreased in the site-based validation, in which the $R^2$ and RMSE values are 0.8 and 14.59 μg/m³, respectively.

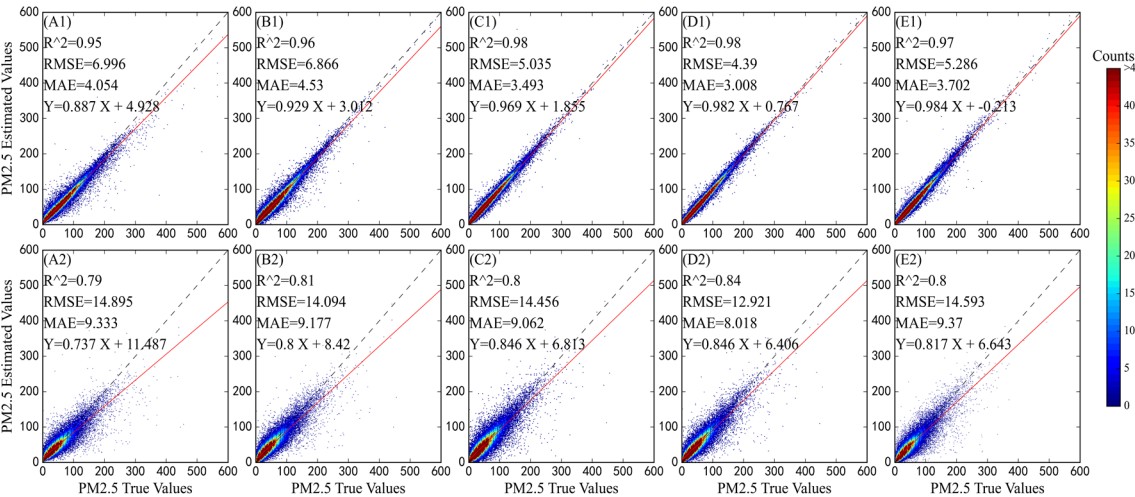

**Figure 3 Accuracy of model Fitting (The first line) and Validation (The second line) (A: RF, B: GBRT, C: DNN, D: RGD-LHMLM (Based on sample), E: RGD-LHMLM (Based on site)). $R^2$ represents determination coefficient, RMSE represents root mean square error, MAE represents mean Absolute Error, N represents the number of samples. The equation Y and X represent the fitting relationship between the actual and estimated PM$_{2.5}$ values. Black dashed line represents 1:1 line, and red line represents best-fit line from linear regression.**

**4.2 Model Performance Analysis**

**4.2.1  Bias analysis of Model**

The average bias of the mixed model in different $PM_{2.5}$ concentration ranges was analyzed, and the result is shown in figure 4. When the $PM_{2.5}$ concentration is less than 60 μg/m³, the average bias of the model is less than 0. As the $PM_{2.5}$ concentration increases, the model deviation gradually increases. In other words, when the $PM_{2.5}$ concentration is small, the predicted value of the model will generally overestimate $PM_{2.5}$, and when the $PM_{2.5}$ further increases, it will underestimate the $PM_{2.5}$ concentration.

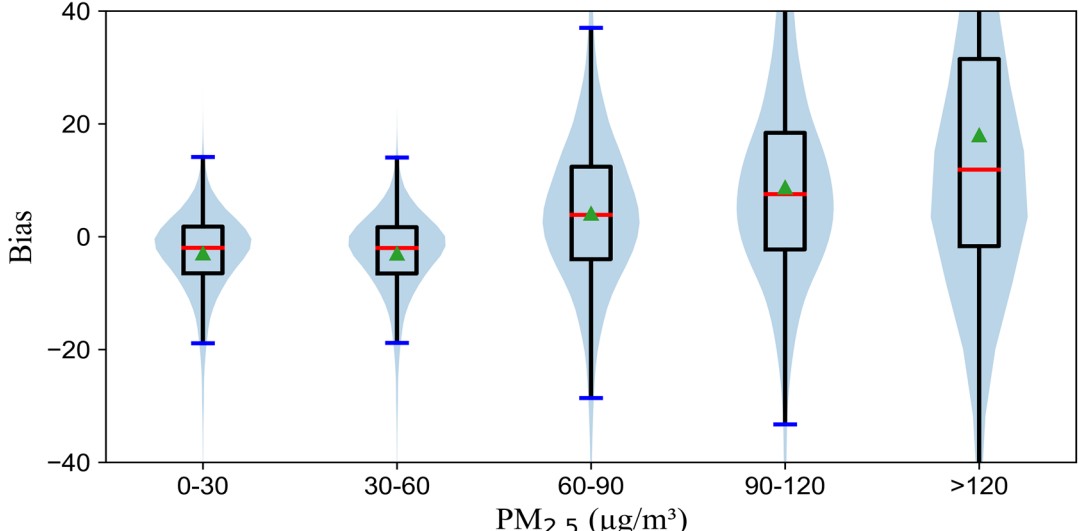

**Figure 4 Boxplots of resulting bias (y-axis) for different $PM_{2.5}$ concentration ranges in μg/m³ (x-axis) (The green arrow symbol, dark blue and red marks represent the average Bias, the median of Bias and the extremum of Bias, respectively. Data density is represented by the light blue shading.)**

**4.2.2 Performance Analysis of Monitoring Station Model**

The spatial performance of the model was analyzed by measuring $R^2$, RMSE, and MAE at the monitoring stations. According to Figure 5, there are regional differences in the inversion performance of RGD-LHMLM. At all monitoring stations, the average $R^2$ was reported 0.74, and $R^2$ was above 0.7 at more than 70% of the stations, especially in the densely populated and industrially developed areas. The model prediction accuracy was reported low ($R^2<0.6$) in Xinjiang, Tibet, Qinghai, Western Sichuan, and a few other areas of Northeast China. The mean values of RMSE and MAE were reported 11.4 μg/m³ and 8.01 μg/m³, respectively. In fact, the mean values of RMSE and MAE were below 20 μg/m³ and 15 μg/m³ in more than 95% of stations, something showed a low estimation error.

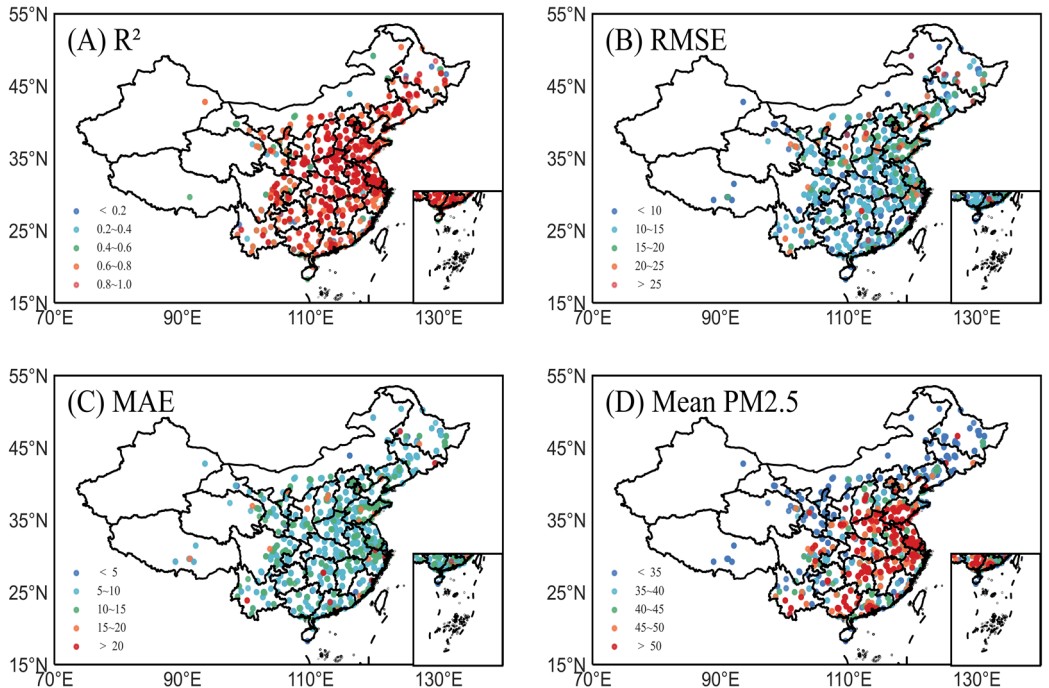

Figure 5 Spatial distributions of model precision in terms of (A) determination coefficient ($R^2$), (B) root mean square error (RMSE), (C) mean Absolute Error (MAE)and (D) mean $PM_{2.5}$ concentration at each site in China. Color circles represent different value ranges of shown statistical parameters.

Based on the analysis of spatial differences in the RGD-LHMLM inversion performance, the following deductions can be made. First, the environmental monitoring stations in the central and eastern regions with better inversion performance were distributed densely, and there are large data available; therefore, the model had a satisfactory training effect. Moreover, data matching was lower in the western region than in other regions, something which resulted in model over-fitting and reduced accuracy (Zhang et al., 2018). Second, some areas of western and northeastern China are covered by snow and the Gobi Desert with high surface albedo. This reduces the accuracy of AOD obtained by satellite observation and brings errors to model training. Finally, the Himawari-8 scanning range is limited, and the satellite observation data obtained in Western China are limited in terms of quantity and accuracy. In general, the RGD-LHMLM has a satisfactory spatial performance, especially in areas with high annual average concentration of $PM_{2.5}$; therefore, it can leave a good inversion effect.

**4.2.3 Time-Scale Model Performance Analysis**

Figure 6 shows the scatterplot fitted with the inversion results of the mixed model from 9:00-17:00 local Time. The model $R^2$ ranged from 0.556 to 0.88 at different times. Except for 17:00 when the model had the worst performance, the model $R^2$ exceeded 0.7 at other times, indicating that the model had a

good performance. The optimal performance time is 13:00, and $R^2$ is 0.88. According to the results, the

hourly differences in model performance were significant.

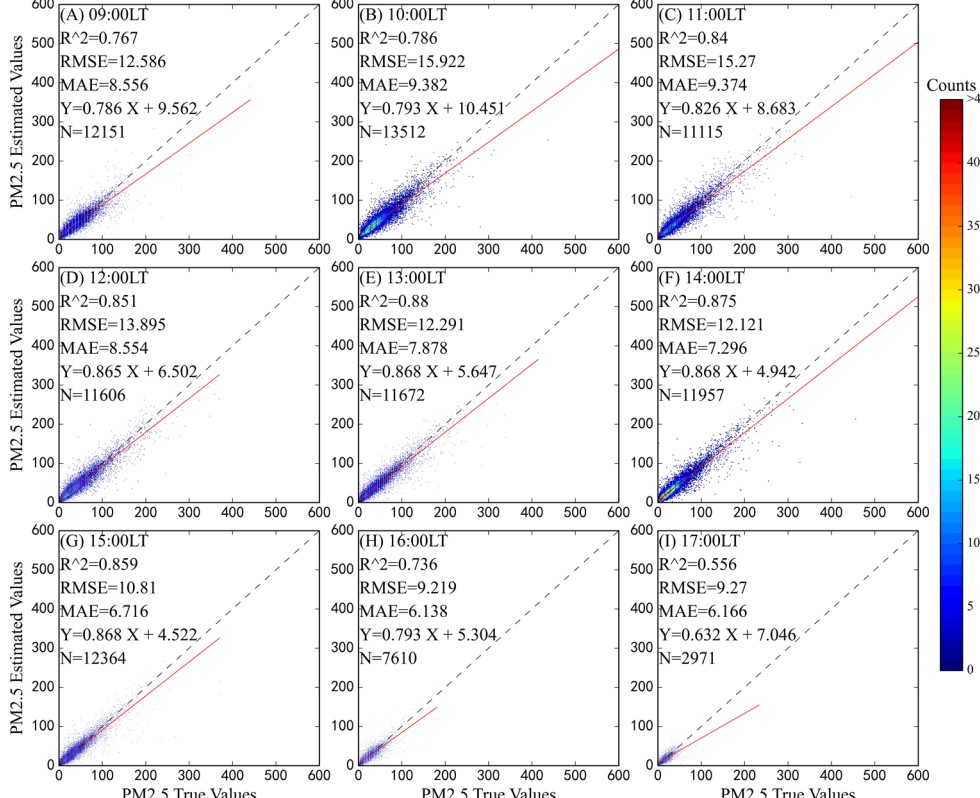

**Figure 6 Density scatterplot of Actual hourly PM$_{2.5}$ values (x-axis) model estimated values (y-axis) in hourly PM$_{2.5}$ estimates in China from (A) 09:00 LT to (I) 17:00 LT. $R^2$ represents determination coefficient, RMSE represents root mean square error, MAE represents mean Absolute Error, N represents the number of samples. The equation Y and X represent the fitting relationship between the actual and estimated PM$_{2.5}$ values. Black dashed line represents 1:1 line, and red line represents best-fit line from linear regression.**

Figure 7 shows the inversion performance results of the hybrid model collected from January to December 2019. The model performed the worst in summer months because $R^2$ was reported 0.73, 0.72, and 0.68, respectively; however, RMSE and MAE were only 9.37, 9.22, 8.26 μg/m$^3$ and 6.59, 6.34, and 5.91 μg/m$^3$, respectively, due to the lower average concentration of PM$_{2.5}$ in summer. Winter and autumn models gained better performance results with an average $R^2$ over 0.8. However, in contrast to summer, the estimation errors of these two seasons were relatively large, with average RMSE of 20.10 μg/m$^3$ and 10.72 μg/m$^3$ and average MAE of 11.20 μg/m$^3$ and 7.25 μg/m$^3$, respectively. The mean $R^2$ was 0.74, whereas the mean RMSE and MAE were 13.71 μg/m$^3$ and 8.39 μg/m$^3$, respectively.

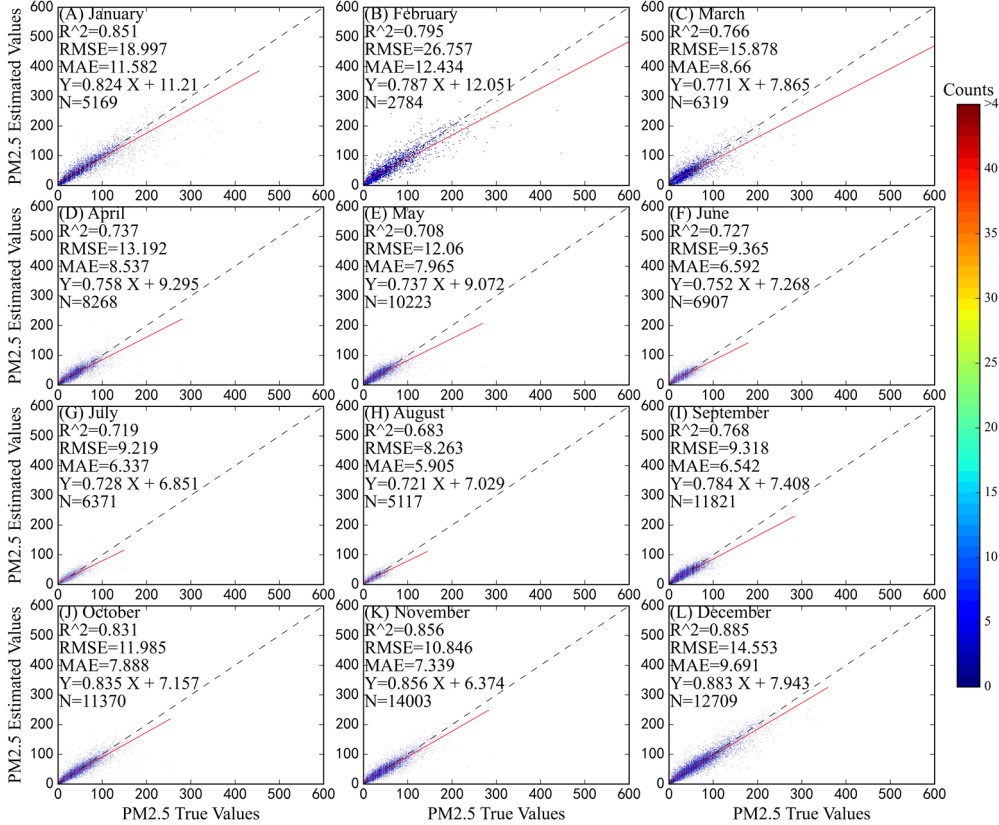

**Figure 7 Same as Figure 6, but for monthly PM₂.₅ estimates.**

### 4.2.4 Feature importance analysis

The model performance differences were also analyzed to extract and rank the model features of RF and GBRT based on the feature importance. The higher the feature importance, the greater the contribution of factors to the model. Figure 8 shows that AOD, boundary layer height, 2 m surface temperature, and relative humidity had the greatest effect on the mixed model performance out of all variable characteristic parameters. Accordingly, AOD is greatly affected by the fine particulate matter and is the main factor in the inversion of $PM_{2.5}$. Changes of the boundary layer height can affect the diffusion ability of the atmosphere. If the boundary layer height is low, the accumulation of pollutants will be caused. At the same time, the 2 m surface temperature has a great impact on the boundary layer height (Miao et al., 2018). Finally, higher rates of atmospheric humidity can improve the fine particulate matter accumulation.

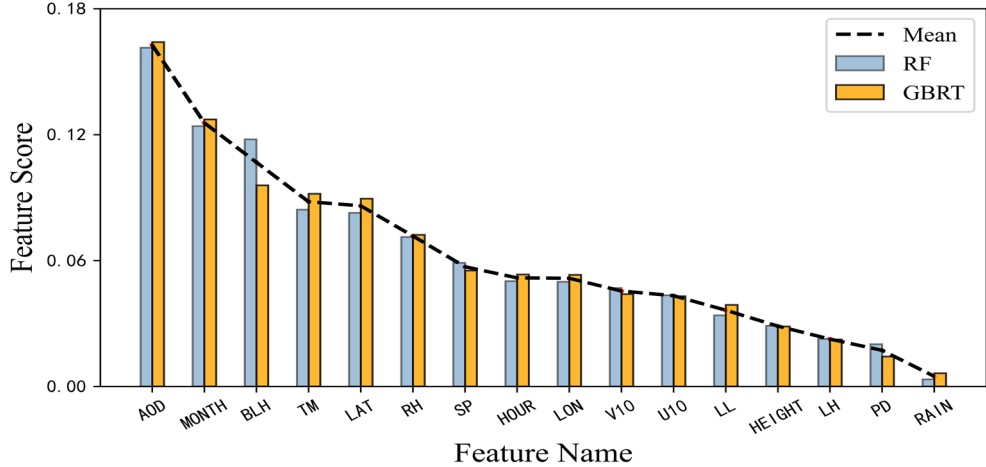

**Figure 8 Score (y-axis) for each model contributing feature factor (y-axis) for the RF (blue) and GBRT (orange). Dashed line represents the mean values.**

The correlation coefficients between the monthly mean values of important meteorological parameters (AOD, BLH, TM and RH) and $R^2$ were also analyzed. According to the results, the correlation coefficients between the meteorological parameters and PM$_{2.5}$ were lower in summer. Furthermore, there are many rainy days and large cloud coverage, which is not conducive to satellite observation and decreases the accuracy of AOD data in summer. Therefore, the summer model performance is poor. There was a strong correlation between meteorological parameters and PM$_{2.5}$ in autumn. There were also similar correlations between spring and winter; however, the winter model performed was better. The reasons can be interpreted as below. The winter temperature and boundary layer height are low, whereas the atmosphere is stable but not conducive to the diffusion of pollutants. Moreover, during the heating period in winter, pollutant emissions soar greatly and result in a sharp rise in the concentration of PM$_{2.5}$. The increased pollution in winter ensures the quality and quantity of data, thereby improving the model performance effectively.

**Table 2 Correlation coefficient between meteorological parameters with PM$_{2.5}$**

| Season | AOD | BLH | TM | RH |
|--------|-----|-----|-----|-----|
| Spring | 0.47 | -0.33 | 0.12 | 0.36 |
| Summer | 0.42 | -0.21 | 0.06 | 0.19 |
| Autumn | 0.38 | -0.29 | 0.24 | 0.41 |
| Winter | 0.44 | -0.33 | 0.12 | 0.35 |

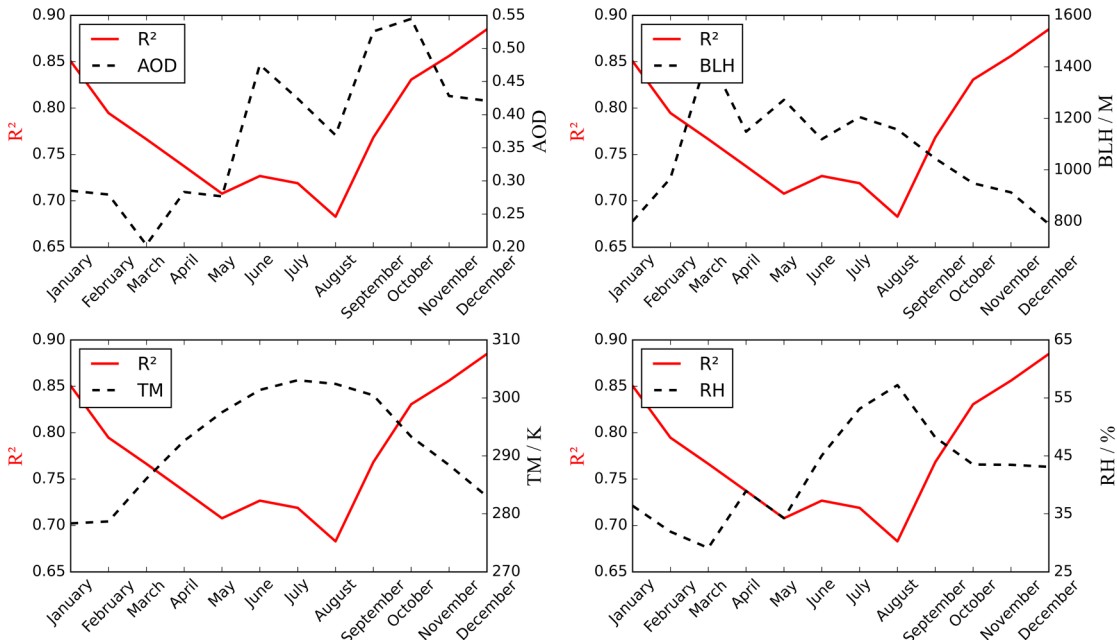

Figure 9 Annual variability (x-axis) of monthly average of meteorological parameters AOD, BLH (m), TM (K), RH (%) (right y-axis) and $R^2$ (left y-axis)

## 4.3 Temporal and Spatial Distribution Characteristics of PM$_{2.5}$ Concentration in China

In terms of spatial distribution, Shandong, Henan, Jiangsu, Anhui, as well as parts of Hubei and Hebei were the most polluted areas in China in 2019, with an annual average PM$_{2.5}$ concentration of 82.86 μg/m$^3$. On the one hand, these areas are economically developed and densely populated, resulting in a large amount of pollutant emissions. On the other hand, the barrier of the peripheral mountains (Taihang Mountains, Qinling Mountains and the Southern Hills) leads to the accumulation of pollutants that are difficult to diffuse. Sichuan Basin is a rare area with a high PM$_{2.5}$ value due to its unique topography (Zhang et al., 2019a), with an annual average PM$_{2.5}$ concentration of 64.69 μg/m$^3$. In addition, Inner Mongolia, Qinghai, Tibet and other places, the pollution level is low, the average annual PM$_{2.5}$ concentration is less than 40 μg/m$^3$.

The temporal distribution of PM$_{2.5}$ is shown in Figure 10, The PM$_{2.5}$ concentration began to rise from 9:00, and peaked at 55.65μg/m$^3$ between 10:00 and 11:00 every day. After that, it maintained a high concentration until 15:00; and began to decrease. In the most polluted areas of China, the peak concentration of PM$_{2.5}$ can reach 85.05μg/m$^3$, while the peak in the less polluted areas is only about 40μg/m$^3$. On a national scale, daily PM$_{2.5}$ concentrations fluctuates slightly.

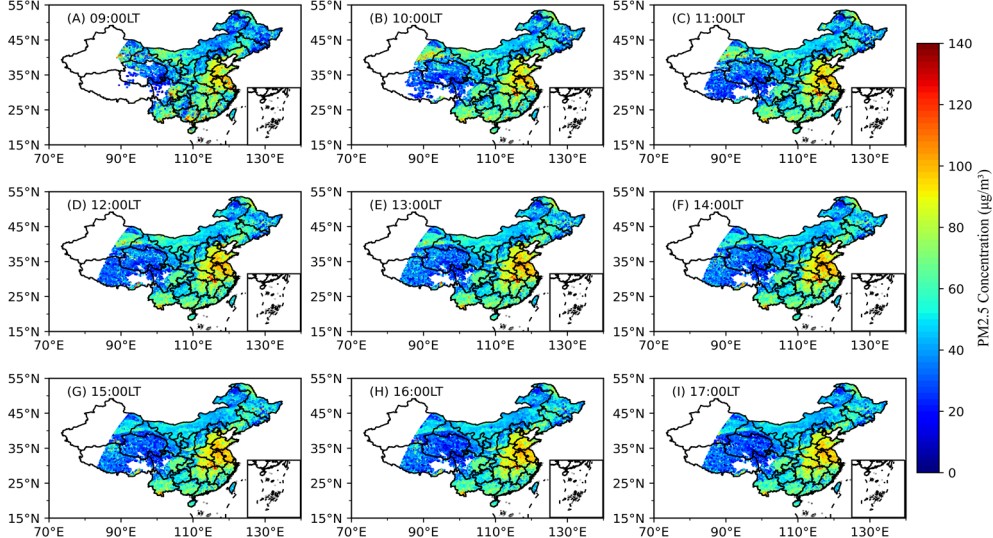

**Figure 10 Hourly Spatial distribution of PM$_{2.5}$ concentration in China at different local times from (A) 09:00**
**LT to (I) 17:00 LT.**

PM$_{2.5}$ concentration in China varies significantly with the seasons. As shown in Figure 11, PM$_{2.5}$

concentration in winter is the highest, with an average value of 62.10μg/m$^3$. January 2019 was the most

polluted month in China, with the average PM$_{2.5}$ concentration reaching 63.58μg/m$^3$. The average PM$_{2.5}$

concentration was 47.39 μg/m$^3$ in summer. The average concentration of PM$_{2.5}$ in spring and autumn was

54.21μg/m$^3$ and 52.26 μg/m$^3$, respectively, indicating similar levels of pollution.

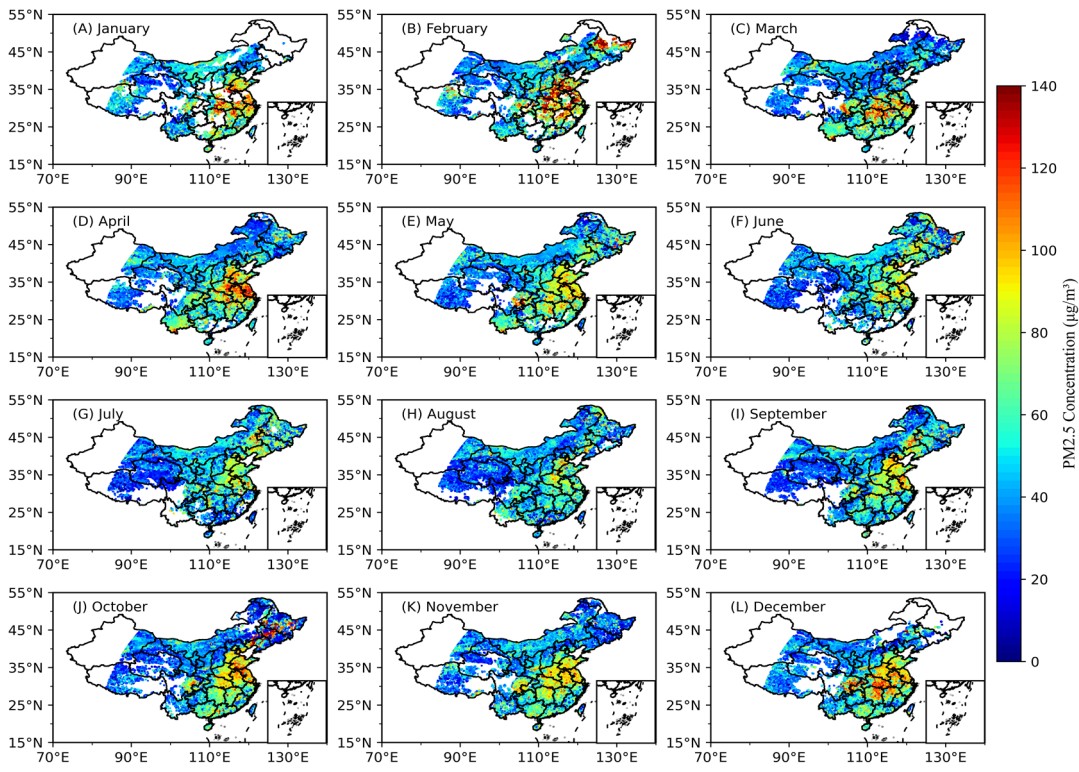

**Figure 11 Same as Fig. 10, but for monthly spatial distribution**

**5 Conclusion**

It is essential to collect the spatiotemporal evolution characteristics regarding the concentration of $PM_{2.5}$ for air pollution prevention and containment. Based on the linear hybrid machine learning model, this paper used the AOD data of Himawari-8 to invert the concentration of $PM_{2.5}$ in China and obtain its distribution characteristics. The model performance and inversion results are analyzed and summarized below:

(1) In the RGD-LHMLM obtained from linear fitting, the DNN accounted for the largest proportion with a weight coefficient of 0.62. The $R^2$ of RGD-LHMLM was 0.84, whereas its generalization ability was significantly better than that of a single model (DNN: 0.80; GBRT: 0.81; RF: 0.79). Moreover, RMSE and MAE were 12.92 $\mu g/m^3$ and 8.01 $\mu g/m^3$, respectively.

(2) The RGD-LHMLM was spatially stable, with $R^2>0.7$ in more than 70% of sites as well as RMSE<20 $\mu g/m^3$ and MAE<15$\mu g/m^3$ in more than 95% of sites. These sites are mainly located in densely populated and industrially developed areas. The correlation difference between the inversion factor and $PM_{2.5}$ in various seasons would lead to seasonal variations in the model performance. In addition, the performance was the worst in summer with an average $R^2$ of 0.71; however, winter showed the best performance with an average $R^2$ of 0.84. The diurnal variation of the model inversion effect is also obvious, and the 11:00-14:00 model usually has better performance.

(3) Changes in the spatiotemporal characteristics were obvious in the concentration of $PM_{2.5}$ in China. In other words, North China and East China had the highest concentration of $PM_{2.5}$ with an average annual concentration of 82.86 $\mu g/m^3$, whereas Inner Mongolia, Qinghai, Tibet, and other regions had low pollution levels with an average annual concentration of $PM_{2.5}$ below 40 $\mu g/m^3$. In winter, the concentration of $PM_{2.5}$ was higher with an average of 62.10 $\mu g/m^3$, whereas the pollution was lighter in summer with an average concentration of $PM_{2.5}$ being reported 47.39 $\mu g/m^3$. In the most polluted areas, the peak concentration of $PM_{2.5}$ can reach 85.05$\mu g/m^3$, but the daily $PM_{2.5}$ concentration fluctuates slightly.

In conclusion, the RGD-LHMLM can accurately measure the concentration of $PM_{2.5}$ and perform the seasonal evolution of pollutants. These results can help control the local pollution. This study also indicated that integrating multiple Machine learning models improved the accuracy of fitting results effectively. For more accurate pollutant data, such models can be employed to fit the $PM_{2.5}$ in the future

with more parameters closely related to $PM_{2.5}$. However, there are some vacant values in the results of this study. There are also no data for some areas. Thus, other satellite data can be used in future studies to solve this problem.

**Code/Data availability**

Datasets and Code related to this paper can be requested from the corresponding author (chenbin@lzu.edu.cn). The $PM_{2.5}$ data download address is : http://106.37.208.233:20035/; Himawari-8 AOD data provided by the Japan Meteorological Agency, download from: http://www.eorc.jaxa.jp/ptree/index.html; ERA-5 meteorological data can be downloaded from the European Centre for Medium-Range Weather Forecasts (ECMWF) at : https://cds.climate.copernicus.eu; Ground elevation SRTM3 data download address is: http://srtm.csi.cgiar.org/index.asp; NASA's social and economic data and the population density of data center, download address is: http://sedac.ciesin.columbia.edu/data/collection/gpw-v4/documentation.

**Author contributions**

Chen proposed the content of the study. Song performed data processing, model building, result analysis, and article writing. Huang, Dong and Yang checked the content of the article.

**Competing interests**

The authors declare that they have no conflict of interest.

**Acknowledgments**

We thank China National Environmental Monitoring Center, Japan Meteorological Agency, European Centre for Medium-Range Weather Forecasts, NASA, and the National Mapping Service of the Department of Defense.

**Financial support**

The National Key Research and Development Program of China (Grant number 2019YFA0606800), the

National Natural Science Foundation of China (Grant 41775021), The Fundamental Research Funds for the Central Universities (Grant lzujbky-2019-43).

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
