# Peer review of "Estimation of PM2.5 Concentration in China Using"

_Atmospheric Measurement Techniques, 2021_

## Referee Comment (RC2)

Anonymous Reviewer

May 21, 2021

The study by Song et al. presents a linear hybrid machine learning model to estimate regional PM$_{2.5}$ distributions from Himawari-8 AOD observations. In the manuscript, the authors stated that the proposed RGD-LHMLM method outperforms than three conventional machine learning methods and can perform accurate estimations.

The topic fits well to the aims and scopes of AMT. Machine learning based methods have been widely used to estimate near-surface PM$_{2.5}$ using satellite AOD observations. As the authors stated in the manuscript, there have been a lot of studies about regional estimation of PM$_{2.5}$ over China. However, in my opinion, the presented material in this study do not sufficiently prove that the proposed method is superior to the other three conventional methods and that it can be used to "perform the seasonal evolution of pollutants ", "help control the local pollution", and "fit the PM$_{2.5}$ in the future". I would expect that the functionality of this hybrid method should be logic and provable, which means that the method has indeed learned and generalized mostly from the satellite AOD observations so that it can be used to estimate/predict unexpected PM$_{2.5}$ features in the future. Unfortunately, in the current manuscript, the authors only use the satellite and ground measurements in 2019 and do not specify any reason why the 2019 data can be considered to be representative. Or, is the focus of this study only on investigating spatiotemporal distributions of PM$_{2.5}$ concentration during 2019? If so, the scientific meaning should be addressed.

In Sections 2 and 3, information about training data quality and its error propagation, and details of the mixed model are missing. Without these details, I cannot justify the model performance. The reasons are given in Section 1.

Therefore, I would not recommend a publication based on the current manuscript. Besides, I do have a number of concerns that require feedback (see Section 1 below).

**1 Specific comments**

- The paper does not provide enough evidence to support the major conclusions. The proposed method does not have generality in terms of target period as the training relies fully on the Himawari-8 AOD data over 2019. What about for the $PM_{2.5}$ estimation in some other years? To have a completely new training? Since the authors did not perform any $PM_{2.5}$ estimation for other years, I'd like to ask whether the training data already includes all possible cases between satellite AOD and ground $PM_{2.5}$. Even if by including more satellite AOD datasets over a longer period, it can still be questionable whether the selected training data are considered to be representative.

- Section 2: Please include information about data quality of all datasets used for training (e.g., satellite AOD, ground-based data, meteorological data). The current training assumes that Himawari-8 AOD and ground $PM_{2.5}$ data are true values, which in reality, is not true. Thus, please discuss how much impact of their data quality on the model performance in a quantitative way, i.e., what is the error propagation of these training data?

- These machine learning based models are sort of "black boxes", which means that it would seem unclear what a physical relationship between input and output are learned, particularly to readers who are not familiar with $PM_{2.5}$ estimation. I would suggest to reformulate the beginning of Section 3 by adding mathematical explanation for such context.

- Section 3: Please specify explicitly the input/output of the training(s).

- Section 3: Please describe in detail the linear combination of the three optimal sub-models.

- Page 8, Line 13: According to Table 1, I do not notice any "significant" improvement from an individual sub-model to a linear-mixed model. I would prefer to say slightly improved, as can be seen also from Figure 3.

- Section 4: The current manuscript only discusses the monthly performance of the linear-mixed model. But as far as I know, the usage of geostationary data such as Himawari-8, is especially beneficial to improving the understanding of daily variation of $PM_{2.5}$. If this study focuses solely on the monthly/seasonal variation, why not use MODIS AOD data over a longer period?

- Figure 5: It seems that the estimated $PM_{2.5}$ are in general lower than the "true" values. Is this underestimation pattern related to Himawari-8 data? Please expand the relevant discussion.

- Figure 6: Please include importance of input parameters to DNN as well.

- Section 4: An error characterization of model estimation is missing. Please discuss (quantitatively if possible) error contributions of the input parameters (at least including dominant error sources) to the final output.

- Page 15, Line 19: Any examples of "other satellite data"? If other satellite observations are considered, how do you optimize the model training, as the current training is only based on Himawari data.

---

## Referee Comment (RC3)

The authors presented a new perspective to derive hourly PM2.5 concentrations from Himawari-8 satellite in China by combing different AI methods. This study is overall good, and the results are generally well presented. However, I still have some concerns and suggestions for the authors to improve the manuscript.

Major comments:
My first concern is that the authors used all the data samples collected at the same locations having ground-based measurements using the cross-validation method, but the PM2.5 predictions are not evaluated at locations where ground-based measurements are unavailable. Thus, I suggest adding an additional validation to test the spatial prediction ability of your model based on the monitoring stations using the cross-validation method.

My other concern is that the purpose of this study is to derive hourly PM2.5 concentrations from geostationary satellites. However, the spatial analysis is performed on a monthly scale (Section 4.3), which will largely reduce the sense of the current study. Thus, it is suggested to add more analysis on PM diurnal variations across China.

Minor comments:
Introduction:
The authors are suggested to update the literature by summarizing more recent studies on PM2.5 estimations using sun-synchronous and geostationary orbit satellites, especially those focusing on the whole of China. Below references may help you found more information on various recent studies to help enrich your study.
https://doi.org/10.1016/j.rse.2020.112136
https://doi.org/10.5194/acp-21-7863-2021

Data:
Section 2.2: Line 15, Reference for Himawari-8 aerosol algorithm is needed.
https://doi.org/10.2151/jmsj.2018-039

Line 17: Below references provide a more comprehensive evaluation of Himawari-8 aerosol products in China.
https://doi.org/10.1016/j.scitotenv.2019.07.326
https://doi.org/10.1016/j.atmosenv.2018.11.024

Section 2.3: Reference for ERA5 reanalysis is needed.

Method:
References for these traditional ML or DL methods are needed, e.g.,
Friedman, J.: Greedy function approximation: a gradient boosting machine, Ann. Stat., 29, 1189–1232, 2001.
Breiman, L.: Random forests, Mach. Learn., 45, 5–32, 2001.

Results and discussion:

Lines 5-9: It is not clear to me how to determine the weight coefficients, and could you add more descriptions?

Section 4.2.2: How about the accuracy of PM2.5 estimations for different hours?

Page 11, Lines 12-15, Page 12, and Page 13, Lines 1-4: May move to a new separate Discussion section.

How about your model compared with those developed in previous studies using the Himawari-8 AOD products in China?

---

## Author Comment (AC1)

Review result of "Estimation of PM2.5 Concentration in China Using Linear Hybrid Machine Learning Model." (AMT-2021-64) by Song et al.

Response to RC1:

referee's comments are given in blue, our responses are given in red.

RC1: The submitted article develops a method to estimate PM2.5 values over China using a linear combination of three machine learning model. The innovative of this approach is the method to have an ensemble PM2.5 data from multiple machine learning model outputs. The research method is solid, and the results are convincing.

Response: We would like to thank the editor and referee for carefully reading the manuscript and providing detailed and constructive comments, which have helped a lot in improving the manuscript. We quote each comment below, followed by our response.

RC1: The background of the research does not cover all of the most recent machine learning produced PM2.5 products over China and provide convincing reason of why this approach is superior to the rest products. The big advantage of using AHI is the high temporal data (sub-hourly), however, the results section does not reflect this advantage.

Response: Due to the early start of this study, the latest research progress

was not quoted when writing the research background. To make up for these deficiencies, we will add 18 references to the manuscript. These references are listed at the page 8-10 of this document.

The advantage that AHI can provide high temporal resolution data is also discussed, but for some reasons it was not included in the previous version of the manuscript. In the revised manuscript we have added this content. The results are shown in the figure below.

Figure 6 shows the scatterplot fitted with the inversion results of the mixed model from 9:00-17:00 Local Time. The model  $R^2$  ranged from 0.556 to 0.88 at different times. Except for 17:00 when the model had the worst performance, the model  $R^2$  exceeded 0.7 at other times, indicating that the model had a good performance. The optimal performance time is 13:00,  $R^2$  is 0.88. According to the results, the hourly differences in model performance were significant.

Figure 6 Hourly validation of model performance

The temporal distribution of  $PM_{2.5}$  is shown in Figure 10, The  $PM_{2.5}$  concentration began to rise from 9:00, and peaked at 55.65µg/m3 between 10:00 and 11:00 every day. After that, it maintained a high concentration until 15:00, and began to decrease. In the most polluted areas of China, the peak concentration of  $PM_{2.5}$  can reach  $85.05\mu$ g/m3, while the peak in the less polluted areas is only about  $40\mu$ g/m3. On a national scale, daily  $PM_{2.5}$  concentrations fluctuate little.

---

## Author Comment (AC2)

Response to RC2:

referee's comments are given in blue,

our responses are given in red.

RC2: The study by Song et al. presents a linear hybrid machine learning model to estimate regional $PM_{2.5}$ distributions from Himawari-8 AOD observations. In the manuscript, the authors stated that the proposed RGD-LHMLM method outperforms than three conventional machine learning methods and can perform accurate estimations.

Response: We would like to thank the editor and referee for carefully reading the manuscript and providing detailed and constructive comments, which have helped a lot in improving the manuscript. We quote each comment below, followed by our response.

RC2: The paper does not provide enough evidence to support the major conclusions. The proposed method does not have generality in terms of target period as the training relies fully on the Himawari-8 AOD data over 2019. What about for the $PM_{2.5}$ estimation in some other years? To have a completely new training? Since the authors did not perform any $PM_{2.5}$ estimation for other years, I'd like to ask whether the training data already includes all possible cases between satellite AOD and ground

*PM$_{2.5}$. Even if by including more satellite AOD datasets over a longer period, it can still be questionable whether the selected training data are considered to be representative.*

Response: Our research is mainly based on two decision tree models and a neural network model to build a semi-explanatory estimation model. This semi-explanatory nature is mainly reflected in the analysis of the feature importance. In other words, deep learning models are often seen as black boxes with low interpretability. Therefore, we want to use the feature importance obtained by the decision tree model and the computational power of deep learning to build the semi-explanatory estimation model. Since DNN has the highest weight coefficient in the final hybrid model, we believe that this assumption has been realized to a certain extent.

Given factors such as climate change and human controls, the data from just one year cannot represent all possible scenarios between AOD and PM$_{2.5}$. However, the monthly and hourly variations contained in the data are very significant, and the number of samples retrieved from this data also meets the requirements of machine learning. So, we believe that one year's datasets can provide better training for the model; on the other hand, the Himawari-8 data was updated when we started this study. Based on the core thesis of this research and the above two reasons, we have selected the Himawari-8 AOD of 2019 for training.

In future research, we will extend the time period to study the change trend of PM$_{2.5}$ on a long time scale.
* * *
*RC2: Section 2: Please include information about data quality of all datasets used for training (e.g., satellite AOD, ground-based data, meteorological data). The current training assumes that Himawari-8 AOD and ground PM$_{2.5}$ data are true values, which in reality, is not true. Thus, please discuss how much impact of their data quality on the model performance in a quantitative way, i.e., what is the error propagation of these training data?*

Response: Ground PM2.5 can be observed by two methods. The first is an automatic analysis method including trace element oscillation balance method or β-ray attenuation method. The other is manual gravimetric method (HJ618). The observed data are calibrated and quality-controlled according to national standards GB 3095-2012 (China's National Ambient air quality standards)(China, 2012).

Himawari-8 AOD is obtained by an aerosol retrieval algorithm based on Lambertian-surface-assumed developed by Yoshida et al. (2018). Himawari-8 AOD was compared with the AOD data of AERONET (Aerosol Robotic Network)(Zhang et al., 2019), the results show that they are consistent (R$^2$=0.75), RMSE and MAE were 0.39 and 0.21, respectively(Wei et al., 2019). In the study, we selected AOD with strict cloud screening, that is, AOD data with low uncertainty.

Uncertainty estimation of ERA5 data has described in detail in the following website: https://confluence.ecmwf.int/display/CKB/ERA5%3A+uncertainty+estimation.

To sum up, the data we used have been quality-controlled and can represent the real situation to some extent. As commented by Referee #2, we have added bias analysis.

There is an irretrievable error between the AOD or $PM_{2.5}$ and its true value. As shown in figure 4, the average bias of the mixed model in different $PM_{2.5}$ concentration ranges was analyzed, and the result is shown in the figure 4. when the $PM_{2.5}$ concentration is less than 60 $\mu g/m^3$, the average bias of the model is less than 0. As the $PM_{2.5}$ concentration increases, the model deviation gradually increases. In other words, when the $PM_{2.5}$ concentration is small, the predicted value of the model will generally overestimate $PM_{2.5}$, and when the $PM_{2.5}$ further increases, it will underestimate the $PM_{2.5}$ concentration.

[Figure]

Figure 4 Bias between model predicted values and label values

In the machine learning algorithm, the error of the model will be corrected continuously according to the label value during the training. As is known to all, the data calculated by the model are mainly related to the factors with high feature importance. In this model, the factor with the highest importance of feature is AOD. That is to say, when there is data error in AOD, it will be transmitted to the forecast result, and when there is data error in $PM_{2.5}$, it will interfere with the error correction of the model. Based on the above discussion, we believe that the errors in the model are mainly caused by the errors of AOD and $PM_{2.5}$ when the pollution is relatively serious. In the case of low $PM_{2.5}$ concentration, this error transfer phenomenon is relatively less.

*RC2: These machine learning based models are sort of "black boxes", which means that it would seem unclear what a physical relationship between input and output are learned, particularly to readers who are not familiar with PM2:5 estimation. I would suggest to reformulate the beginning of Section 3 by adding mathematical explanation for such context.*

Response: It is a good suggestion. We will add the mathematical expression of the sub-model in the revised manuscript.

$$PM_{2.5i,j} = AOD_{i,j} + BLH_{i,j} + RH_{i,j} + TM_{i,j} + LL_{i,j} + LH_{i,j} + SP_{i,j} \quad (1)$$
$$+ RAIN_{i,j} + U_{10i,j} + V_{10i,j} + PD_{i,j} + HEIGHT_{i,j} + LON_{i,j}$$
$$+ LAT_{i,j} + MONTH_{i,j} + HOUR_{i,j}$$

Formula (1) is applicable to RF, GBRT and DNN. Where $PM_{2.5i,j}$ is the PM$_{2.5}$ at time i on station j.

*RC2: Section 3: Please specify explicitly the input/output of the training(s).*

Response: The input is 16 features including AOD (aerosol optical depth), surface relative humidity (RH, expressed as a percentage), air temperature at a height of 2 m (TM, expressed as K), Wind speed (U10, V10, in m/s), surface pressure (SP, in Pa), boundary layer height (BLH, in m) and cumulative precipitation (RAIN, in m) at 10 m above the ground, high and low vegetation index (LH, LL), ground elevation data (DEM), population density data (PD), longitude, latitude, month and hour.

The output is PM2.5 concentrations.

*RC2: Section 3: Please describe in detail the linear combination of the three optimal sub-models.*

Response: The coefficient is determined by multiple linear regression model. Firstly, we use three sub-models to calculate the predicted value under the corresponding model. Then, multiple linear regressions are performed between the calculated predicted values and the label values in the original data. Finally, the output coefficients and intercepts of the multiple linear regression model are taken as the parameters of the **RGD-LHMLM**.

*RC2: Page 8, Line 13: According to Table 1, I do not notice any*

*"significant" improvement from an individual sub-model to a linear-mixed model. I would prefer to say slightly improved, as can be seen also from Figure 3.*

Response: We have revised the description in the revised manuscript.
* * *
*RC2: Section 4: The current manuscript only discusses the monthly performance of the linear-mixed model. But as far as I know, the usage of geostationary data such as Himawari-8, is especially beneficial to improving the understanding of daily variation of $PM_{2.5}$. If this study focuses solely on the monthly/seasonal variation, why not use MODIS AOD data over a longer period?*

The advantage that AHI can provide high temporal resolution data is also discussed, but for some reasons it was not included in the previous version of the manuscript. In the revised manuscript we have added this content. The results are shown in the figure below.

Figure 6 shows the scatterplot fitted with the inversion results of the mixed model from 9:00-17:00 Local Time. The model $R^2$ ranged from 0.556 to 0.88 at different times. Except for 17:00 when the model had the worst performance, the model $R^2$ exceeded 0.7 at other times, indicating that the model had a good performance. The optimal performance time is 13:00, $R^2$ is 0.88. According to the results, the hourly differences in model performance were significant.

[Figure]

Figure 6 Hourly validation of model performance

The temporal distribution of PM$_{2.5}$ is shown in Figure 10, The PM$_{2.5}$ concentration began to rise from 9:00, and peaked at 55.65μg/m3 between 10:00 and 11:00 every day. After that, it maintained a high concentration until 15:00, and began to decrease. In the most polluted areas of China, the peak concentration of PM$_{2.5}$ can reach 85.05μg/m$^3$, while the peak in the less polluted areas is only about 40μg/m$^3$. On a national scale, daily PM$_{2.5}$ concentrations fluctuate little.

[Figure]

Figure 10 Hourly distribution of PM₂.₅ in China in 2019
* * *
*RC3: Figure 5: It seems that the estimated PM2.5 are in general lower than the "true" values. Is this underestimation pattern related to Himawari-8 data? Please expand the relevant discussion.*

Response: That's a very good question. As we all know, AOD is the integral of the aerosol extinction coefficient from the surface to the top of the atmosphere, and PM2.5 is small aerosol particles close to the surface which could float in the atmosphere for long period. Thus, PM2.5 contributes a significant portion of AOD, and the correlation between AOD and PM2.5 has a strong spatial and temporal variation(Ma et al., 2016;Xu et al., 2021). Combined with the feature importance of AOD and the above content, We believe that AOD has a very important influence on the model prediction values. In some studies,

however, Himawari-8 AOD has been found to be underestimated(Zang et al., 2018). Therefore, we believe that the underestimation of PM2.5 is closely related to the value of AOD. But, we need to note that the impact of meteorological parameters on the relationship between PM2.5 and AOD cannot be ignored (Gupta et al., 2006). So, the underestimated PM2.5 predicted value is greatly related to the influence of AOD, but the influence of meteorological factors should also be considered.

RC3: Figure 6: Please include importance of input parameters to DNN as well.

Response: As is answered in the first question, the feature importance of deep learning is difficult to obtain, and we only use the strong computational power of DNN to build the model. The DNN input is the same as the tree model, and the importance of the features in the tree model can explain which features are more important. In future research, we will study how to obtain the feature importance of DNN, and isolate them for analysis.

RC3: Section 4: An error characterization of model estimation is missing. Please discuss (quantitatively if possible) error contributions of the input parameters (at least including dominant error sources) to the final output.

Response: That's a tremendously good suggestion. We believe that the greater the importance of a feature in a model, the greater its contribution to the error of the model when there is an error. Perhaps this is not a

sufficient explanation. In future studies, we will try to discuss the error contribution of input parameters to the model.
* * *
*RC2: Page 15, Line 19: Any examples of "other satellite data"? If other satellite observations are considered, how do you optimize the model training, as the current training is only based on Himawari-8 data.*

Response: Some studies used "other satellite data", such as FY-4A(Mao et al., 2021), MODIS(Wei et al., 2021b), GOIC(Tang et al., 2019) and VIIRS(Yao et al., 2019).

*"If other satellite are considered"*, I have two understandings. If it means not using Himawari-8 AOD data but using other satellite data for training, then the optimization process of the model is no different with Himawari-8. If this means using both Himawari-8 AOD data and other satellite data for training, then I think it's best to merge the two AOD datasets. In other words, the two kinds of AOD data are unified into one kind of integrated AOD data through linear regression or other algorithms. There are two benefits to doing this: firstly, The integrated AOD data can improve the data coverage to the surface; secondly, reducing the number of features can reduce the training time of the model and improve the efficiency.

We fully agree with the Referee #2's opinion, and our follow-up work will be done through multi-satellite data fusion.

We have compared other studies with our own and listed the results in Table 1:

Table 1

| Model | $R^2$ | RMSE | MAE | Reference |
|---|---|---|---|---|
| Stacking model | 0.85 | 17.3 | 10.5 | (Chen et al., 2019) |
| Two-stage random forests (YRD) | 0.86 | 12.4 | / | (Tang et al., 2019) |
| LME (BTH) | 0.86 | 24.5 | 14.2 | (Wang et al., 2017) |
| GTWR | 0.78 | 20.10 | / | (Xue et al., 2020) |
| STLG | 0.85 | 13.62 | 8.49 | (Wei et al., 2021a) |
| RGD-LHMLM | 0.84 | 12.92 | 8.01 | This paper |

**reference**

Chen, J. P., Yin, J. H., Zang, L., Zhang, T. X., and Zhao, M. D.: Stacking machine learning model for estimating hourly PM2.5 in China based on Himawari 8 aerosol optical depth data,Sci Total Environ, 697,https://doi.org/10.1016/j.scitotenv.2019.134021, 2019.

China: Ambient air quality standards. GB 3095-2012., China Environmental Science Press, Beijing, 2012.

Gupta, P., Christopher, S. A., Wang, J., Gehrig, R., Lee, Y., and Kumar, N.: Satellite remote sensing of particulate matter and air quality assessment over global cities,Atmos Environ, 40, 5880-5892,https://doi.org/10.1016/j.atmosenv.2006.03.016, 2006.

Ma, X. Y., Wang, J. Y., Yu, F. Q., Jia, H. L., and Hu, Y. N.: Can MODIS AOD be employed to derive PM2.5 in Beijing-Tianjin-Hebei over China?,Atmos Res, 181, 250-256,https://doi.org/10.1016/j.atmosres.2016.06.018, 2016.

Mao, F., Hong, J., Min, Q., Gong, W., Zang, L., and Yin, J.: Estimating hourly full-coverage PM2.5 over China based on TOA reflectance data from the Fengyun-4A satellite,Environ Pollut, 270, 116119,https://doi.org/10.1016/j.envpol.2020.116119, 2021.

Tang, D., Liu, D. R., Tang, Y. L., Seyler, B. C., Deng, X. F., and Zhan, Y.: Comparison of GOCI and Himawari-8 aerosol optical depth for deriving full-coverage hourly PM2.5 across the Yangtze River Delta,Atmos Environ, 217,https://doi.org/10.1016/j.atmosenv.2019.116973, 2019.

Wang, W., Mao, F. Y., Du, L., Pan, Z. X., Gong, W., and Fang, S. H.: Deriving Hourly PM2.5 Concentrations from Himawari-8 AODs over Beijing-Tianjin-Hebei in China,Remote Sens-Basel, 9,https://doi.org/10.3390/rs9080858, 2017.

Wei, J., Li, Z., Sun, L., Peng, Y., Zhang, Z., Li, Z., Su, T., Feng, L., Cai, Z., and Wu, H.: Evaluation and uncertainty estimate of next-generation geostationary meteorological Himawari-8/AHI aerosol products,Sci Total Environ, 692, 879-891,https://doi.org/10.1016/j.scitotenv.2019.07.326, 2019.

Wei, J., Li, Z., Pinker, R. T., Wang, J., Sun, L., Xue, W., Li, R., and Cribb, M.: Himawari-8-derived diurnal variations in ground-level PM2.5 pollution across China using the fast space-time Light Gradient Boosting Machine (LightGBM),Atmos. Chem. Phys., 21, 7863-7880,https://doi.org/10.5194/acp-21-7863-2021, 2021a.

Wei, J., Li, Z. Q., Lyapustin, A., Sun, L., Peng, Y. R., Xue, W. H., Su, T. N., and Cribb, M.: Reconstructing 1-km-resolution high-quality PM2.5 data records from 2000 to 2018 in China: spatiotemporal variations and policy implications,Remote Sens Environ, 252,https://doi.org/10.1016/j.rse.2020.112136, 2021b.

Xu, Q. Q., Chen, X. L., Yang, S. B., Tang, L. L., and Dong, J. D.: Spatiotemporal relationship between Himawari-8 hourly columnar aerosol optical depth (AOD) and ground-level PM2.5 mass concentration in mainland China,Sci Total Environ, 765,https://doi.org/10.1016/j.scitotenv.2020.144241, 2021.

Xue, Y., Li, Y., Guang, J., Tugui, A., She, L., Qin, K., Fan, C., Che, Y. H., Xie, Y. Q., Wen, Y. N., and Wang, Z. X.: Hourly PM2.5 Estimation over Central and Eastern China Based on Himawari-8 Data,Remote Sens-Basel, 12,https://doi.org/10.3390/rs12050855, 2020.

Yao, F., Wu, J., Li, W., and Peng, J.: A spatially structured adaptive two-stage model for retrieving ground-level PM2.5 concentrations from VIIRS AOD in China,ISPRS Journal of Photogrammetry and Remote Sensing, 151, 263-276,https://doi.org/10.1016/j.isprsjprs.2019.03.011, 2019.

Yoshida, M., Kikuchi, M., Nagao, T. M., Murakami, H., Nomaki, T., and Higurashi, A.: Common Retrieval of Aerosol Properties for Imaging Satellite Sensors,Journal of the Meteorological Society of

Japan. Ser. II, 96B, 193-209,https://doi.org/10.2151/jmsj.2018-039, 2018.

Zang, L., Mao, F., Guo, J., Gong, W., Wang, W., and Pan, Z.: Estimating hourly PM1 concentrations from Himawari-8 aerosol optical depth in China,Environ Pollut, 241, 654-663,https://doi.org/10.1016/j.envpol.2018.05.100, 2018.

Zhang, Z., Wu, W., Fan, M., Tao, M., Wei, J., Jin, J., Tan, Y., and Wang, Q.: Validation of Himawari-8 aerosol optical depth retrievals over China,Atmos Environ, 199, 32-44,https://doi.org/10.1016/j.atmosenv.2018.11.024, 2019.

---

## Author Comment (AC3)

Review result of "Estimation of PM$_{2.5}$ Concentration in China Using Linear Hybrid Machine Learning Model." (AMT-2021-64) by Song et al.

Response to RC3:

referee's comments are given in blue,

our responses are given in red.

*RC3: The authors presented a new perspective to derive hourly PM2.5 concentrations from Himawari-8 satellite in China by combing different AI methods. This study is overall good, and the results are generally well presented.*

Response: We would like to thank the editor and referee for carefully reading the manuscript and providing detailed and constructive comments, which have helped a lot in improving the manuscript. We quote each comment below, followed by our response.
* * *
*RC3: My first concern is that the authors used all the data samples collected at the same locations having ground-based measurements using the cross-validation method, but the PM$_{2.5}$ predictions are not evaluated at locations where ground-based measurements are unavailable. Thus, I suggest adding an additional validation to test the spatial prediction ability of your model based on the monitoring stations using the cross-validation method.*

Response: We strongly agree with the comment. We have added the additional validation based on the monitoring stations. The results are shown in Fig. 3 (E), with a decrease in accuracy. In future studies, therefore, we should add better spatial predictor features.

[Figure]

Figure 3 Accuracy of model Fitting and Validation (A: RF, B: GBRT, C: DNN, D: RGD-LHMLM (Based on sample), E: RGD-LHMLM (Based on site))

*RC3: My other concern is that the purpose of this study is to derive hourly PM2.5 concentrations from geostationary satellites. However, the spatial analysis is performed on a monthly scale (Section 4.3), which will largely reduce the sense of the current study. Thus, it is suggested to add more analysis on PM diurnal variations across China.*

The advantage that AHI can provide high temporal resolution data is also discussed, but for some reasons it was not included in the previous version of the manuscript. In the revised manuscript we have added this content. The results are shown in the figure below.

Figure 6 shows the scatterplot fitted with the inversion results of the mixed model from 9:00-17:00 Local Time. The model $R^2$ ranged from 0.556 to 0.88 at different times. Except for 17:00 when the model had the worst performance, the model $R^2$ exceeded 0.7 at other times, indicating that the model had a good performance. The optimal performance time is 13:00, $R^2$ is 0.88. According to the results, the hourly differences in model performance were significant.

[Figure]

Figure 6 Hourly validation of model performance

The temporal distribution of $PM_{2.5}$ is shown in Figure 10, The $PM_{2.5}$ concentration began to rise from 9:00, and peaked at 55.65μg/m3 between 10:00 and 11:00 every day. After that, it maintained a high concentration until 15:00, and began to decrease. In the most polluted areas of China, the peak concentration of $PM_{2.5}$ can reach 85.05μg/m$^3$, while the peak in the less polluted areas is only about 40μg/m$^3$. On a national scale, daily $PM_{2.5}$ concentrations fluctuate little.

[Figure]

Figure 10 Hourly distribution of PM$_{2.5}$ in China in 2019
* * *
*RC3: The authors are suggested to update the literature by summarizing more recent studies on PM$_{2.5}$ estimations using sun-synchronous and geostationary orbit satellites, especially those focusing on the whole of China. Below references may help you found more information on various recent studies to help enrich your study.*

*Section 2.2: Line 15, Reference for Himawari-8 aerosol algorithm is needed.*

*Line 17: Below references provide a more comprehensive evaluation of Himawari-8 aerosol products in China.*

*Section 2.3: Reference for ERA5 reanalysis is needed.*

*References for these traditional ML or DL methods are needed.*

Response: Many thanks for the references that were provided to our paper.

We have included it in the revised manuscript.
* * *
*RC3: Lines 5-9: It is not clear to me how to determine the weight coefficients, and could you add more descriptions?*

Response: The coefficient is determined by multiple linear regression model. Firstly, we use three sub-models to calculate the predicted value under the corresponding model. Then, multiple linear regressions are performed between the calculated predicted values and the label values in the original data. Finally, the output coefficients and intercepts of the multiple linear regression model are taken as the parameters of the weight coefficients.
* * *
*RC3: Section 4.2.2: How about the accuracy of PM2.5 estimations for different hours?*

Response: Figure 6 shows the scatterplot fitted with the inversion results of the mixed model from 9:00-17:00 Local Time. The model $R^2$ ranged from 0.556 to 0.88 at different times. Except for 17:00 when the model had the worst performance, the model $R^2$ exceeded 0.7 at other times, indicating that the model had a good performance. The optimal performance time is 13:00, $R^2$ is 0.88. According to the results, the hourly differences in model performance were significant.
* * *
*RC3: Page 11, Lines 12-15, Page 12, and Page 13, Lines 1-4: May move to a new separate Discussion section.*

Response: This is a very good comment, and we have adjusted it in the

revised manuscript. The part pointed out by the Referee #3 has been taken as a separate subsection.

*RC3: How about your model compared with those developed in previous studies using the Himawari-8 AOD products in China?*

Response: We have compared other studies with our own and listed the results in Table 1:

Table 1

| Model | $R^2$ | RMSE | MAE | Reference |
|-------|-------|------|-----|-----------|
| Stacking model | 0.85 | 17.3 | 10.5 | (Chen et al., 2019) |
| Two-stage random forests (YRD) | 0.86 | 12.4 | / | (Tang et al., 2019) |
| LME (BTH) | 0.86 | 24.5 | 14.2 | (Wang et al., 2017) |
| GTWR | 0.78 | 20.10 | / | (Xue et al., 2020) |
| STLG | 0.85 | 13.62 | 8.49 | (Wei et al., 2021) |
| RGD-LHMLM | 0.84 | 12.92 | 8.01 | This paper |

According to the result of the table 1, the accuracy of our model is similar to other models, both of which can better complete the estimation of $PM_{2.5}$.

**References**

Chen, J. P., Yin, J. H., Zang, L., Zhang, T. X., and Zhao, M. D.: Stacking machine learning model for estimating hourly PM2.5 in China based on Himawari 8 aerosol optical depth data,Sci Total Environ, 697,https://doi.org/10.1016/j.scitotenv.2019.134021, 2019.

Tang, D., Liu, D. R., Tang, Y. L., Seyler, B. C., Deng, X. F., and Zhan, Y.: Comparison of GOCI and Himawari-8 aerosol optical depth for deriving full-coverage hourly PM2.5 across the Yangtze River Delta,Atmos Environ, 217,https://doi.org/10.1016/j.atmosenv.2019.116973, 2019.

Wang, W., Mao, F. Y., Du, L., Pan, Z. X., Gong, W., and Fang, S. H.: Deriving Hourly PM2.5 Concentrations from Himawari-8 AODs over Beijing-Tianjin-Hebei in China,Remote Sens-Basel,

9,https://doi.org/10.3390/rs9080858, 2017.

Wei, J., Li, Z., Pinker, R. T., Wang, J., Sun, L., Xue, W., Li, R., and Cribb, M.: Himawari-8-derived diurnal variations in ground-level PM2.5 pollution across China using the fast space-time Light Gradient Boosting Machine (LightGBM),Atmos. Chem. Phys., 21, 7863-7880,https://doi.org/10.5194/acp-21-7863-2021, 2021.

Xue, Y., Li, Y., Guang, J., Tugui, A., She, L., Qin, K., Fan, C., Che, Y. H., Xie, Y. Q., Wen, Y. N., and Wang, Z. X.: Hourly PM2.5 Estimation over Central and Eastern China Based on Himawari-8 Data,Remote Sens-Basel, 12,https://doi.org/10.3390/rs12050855, 2020.

---

## Editor Decision (ED1)

Although the paper has significantly improved following the review process, a few remaining technical issues must be address:

Page 3, Line 12-13: The authors should briefly elaborate on the cited Wei et al., (2021a) reference, indicating the Himawari data used and obtained results.

Page 3, Line 30: Abbreviations should be given full names when they first appear in the main text, i.e., space-time random forest (STRF); space-time extra-trees (STET). Please double-check and correct such issues throughout the paper.

Improvements to figure captions are necessary as indicated below

It is suggested that the density scatterplots in Figure 3 (current in a 2X5 multi-panel plot) , be plotted as 5X2. Also, please, clearly indicate in the caption what each axis represents (i.e., density scatter plots of A (x-axis) versus B (y-axis)). Clearly explain what the A1 to E1 and A2 to E2 panels represent using full names for the six locations. Also describe the legend included in the panels.

Figure 4: The y-axis dynamic range needs to be expanded (maybe [-40, 40]?), and the caption should be revised to "Boxplots of resulting bias (y-axis) for different PM2.5 concentration ranges in µg/m3 (x-axis)". Describe in the caption the meaning of the green arrow symbol, dark blue and red marks, and the light blue shading.

Figure 5: Suggested caption:
Spatial distributions of model precision in terms of (A)…, (B)…,(C)…. and (D) mean PM2.5 concentration at each site in China. Use full description of the statistical parameters shown in A, B, and C. Add 'Color circles represent different value ranges of shown statistical parameters.'

Figure 6: Suggested caption:
Density scatterplot of 'Actual hourly PM2.5 values (x-axis) model estimated values (y-axis) in hourly PM2.5 estimates in China from (A) 09:00 LT to (I) 17:00 LT. Blue dashed line represents……., and red line represents …..'. Also, list the statistic parameters of the legends in each panel'

Figure 7: Suggested caption: "Same as Figure 6, but for monthly PM2.5 estimates."

Figure 8: Suggested caption : "Score (y-axis) for each model contributing feature factor (y-axis) for the RF(blue) and GBRT (orange) deep neural networks. Dashed line represents the mean values."

Figure 9. This plot is confusing. It is better using black solid and dashed lines instead of blue and orange colors. Please clearly specify in the plots and in the captions, what is plotted in the left and right y-axes of each of the four plots.  All the axes' titles should be clearly labeled in the figures and described in the figure caption.

Figure 9: Suggested caption: "Annual variability (x-axis) of monthly average of meteorological parameters AOD, BLH (m), TM (K), RH (%) (right y-axis) and R2 (left y-axis)"

Figure 10. Current caption seems wrong.
Suggested caption: "Hourly Spatial distribution of PM2.5 concentration in China at different local times from (A) 09:00 LT to (I) 17:00 LT."

Figure 11. Suggested Caption: "Same as Fig. 10, but for monthly spatial distribution"